# Cold Formabilities of Martensite-Type Medium Mn Steel

**Koh-ichi Sugimoto** [1],*, **Hikaru Tanino** [1] **and Junya Kobayashi** [2]

[1] School of Science and Technology, Shinshu University, Nagano 380-8553, Japan; 14tm116j@shinshu-u.ac.jp
[2] School of Science and Engineering, Ibaraki University, Hitachi 316-8551, Japan; junya.kobayashi.jkoba@vc.ibaraki.ac.jp
* Correspondence: sugimot@shinshu-u.ac.jp; Tel.: +81-90-9667-4482

**Abstract:** Cold stretch-formability and stretch-flangeability of 0.2%C-1.5%Si-5.0%Mn (in mass%) martensite-type medium Mn steel were investigated for automotive applications. High stretch-formability and stretch-flangeability were obtained in the steel subjected to an isothermal transformation process at temperatures between $M_s$ and $M_f - 100\ ^\circ C$. Both formabilities of the steel decreased compared with those of 0.2%C-1.5%Si-1.5Mn and -3Mn steels (equivalent to TRIP-aided martensitic steels), despite a larger or the same uniform and total elongations, especially in the stretch-flangeability. The decreases were mainly caused by the presence of a large amount of martensite/austenite phase, although a large amount of metastable retained austenite made a positive contribution to the formabilities. High Mn content contributed to increasing the stretch-formability.

**Keywords:** stretch-formability; stretch-flangeability; martensite-type medium Mn steel; retained austenite; heat-treatment; isothermal transformation process; direct quenching

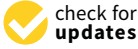



## 1. Introduction

To date, the first, second and third-generation advanced ultrahigh- and high-strength sheet steels (AHSSs) have been developed for the weight reduction and high crush safety of automobiles [1–4]. In most AHSSs, their ductility is enhanced by transformation-induced plasticity (TRIP) [5] and/or twinning-induced plasticity (TWIP) [6] of metastable retained austenite, reverted austenite, and austenite. These AHSSs are categorized as follows [4],

1. First-generation AHSS: ferrite-martensite dual-phase (DP) steel [2,3], TRIP-aided polygonal ferrite (TPF) steel [2], TRIP-aided annealed martensite (TAM) steel [7] and complex-phase (CP) steel [3],
2. Second-generation AHSS: high-Mn TWIP and TWIP/TRIP steels [6],
3. Third-generation AHSS (Type A): TRIP-aided bainitic ferrite (TBF) steel [3,8–10], one-step and two-step quenching and partitioning (Q&P) steels [3,11–13], carbide-free bainitic (CFB) steel [14–16], and duplex-type, laminate-type, and bainitic ferrite-type medium manganese (D–MMn [17–22], L-MMn [23–25], and BF-MMn [26]) steels,
4. Third-generation AHSS (Type B): TRIP-aided martensitic (TM) steel [27–29] and martensite-type medium manganese (M–MMn) steel [30–33].

The second-generation AHSSs are mainly high-Mn austenitic steels with an Mn content higher than 14 mass% and possess the highest ductility by TWIP/TRIP effects. The first- and third-generation AHSSs contain metastable retained austenite or reverted austenite of 5 to 40 vol.%—except for the DP and CP steels—and possess high ductility and cold formability by the TRIP effect of the retained austenite or reverted austenite. The third-generation AHSSs are classified into two types, Type A and Type B, by their kinds of matrix structures and tensile strengths (TS) [4]. The matrix structures and the TSs of Type A are bainitic ferrite (BF) and bainitic ferrite/martensite (BF/M), and higher than 1.0 GPa, respectively, except for D–MMn and L-MMn steels with an annealed martensite [17–25] and/or δ-ferrite [24] matrix structure. On the other hand, the main matrix structure of Type B is martensite and its TS is higher than 1.5 GPa [27–33].

The M–MMn steel contains a larger amount of retained austenite than the TM steel [30–33], although its amount is much less than those of the D–MMn and L-MMn steels [17–25]. Resultantly, the M–MMn steel has an ultra-high tensile strength and shows a large elongation by the TRIP effect of the retained austenite, although the impact toughness is inferior to that of TM steel [30]. If the M–Mn steel achieves higher cold formability than the TM and D–MMn steels, significant weight reduction of the automobiles is expected. Many researchers have reported the cold formability of D–MMn steel because the steel achieves superior formability [22]. However, the cold formability of the M–MMn steel has been hardly investigated to date, although its total elongation is higher than those of TM and D–MMn steels [31].

In this study, the ductility, stretch-formability, and stretch-flangeability of M–MMn steel with a chemical composition of 0.2%C-1.5%Si-5.0%Mn (in mass%) were investigated for applications to the cold-stamping and cold-forging of automotive parts. In addition, these formabilities were related to the microstructural properties. Moreover, the formabilities were compared to the steels with lower Mn content, which are equivalent to TM steels.

## 2. Experimental Procedures

A steel containing 5 mass% Mn was prepared in the form of 100 kg slabs by vacuum melting. To investigate the Mn-content dependencies of microstructure, tensile properties, and formability, the steel slabs, containing low Mn content (1.5 and 3 mass% Mn) were also prepared. Hereafter, the steels with 1.5, 3, and 5 mass% Mn are referred to as 1.5Mn, 3Mn, and 5Mn steels, respectively. The chemical composition, austenite-finish and -start temperatures ($Ac_3$, $Ac_1$ in °C), and martensite-start and -finish temperatures ($M_s$ and $M_f$ in °C) of these slabs are shown in Table 1. The slabs were then heated to 1200 °C and hot-rolled to 5 mm thickness with a finishing temperature of 850 °C, followed by air-cooling to room temperature. After being ground to a thickness of 3 mm, the plates were cold-rolled into sheets of 1.2 mm thickness, with the assistance of annealing, at 650 °C. The CCT diagrams of these steels are shown in Figure 1a. No bainitic transformation appears only in 5Mn steel in a cooling rate range above 0.3 °C/s. The 1.5Mn and 3Mn steels subjected to the DQ process and the IT process at temperatures lower than $M_f$ is equivalent to the TM steel [27–29].

**Table 1.** Chemical composition (mass%) and various transformation temperatures ($Ac_3$, $Ac_1$, $M_s$, and $M_f$ in °C) of steels used.

| Steel | C | Si | Mn | P | S | Al | N | O | $Ac_3$ | $Ac_1$ | $M_s$ | $M_f$ |
|-------|------|------|------|-------|--------|-------|--------|--------|------|------|------|------|
| 1.5Mn | 0.20 | 1.49 | 1.50 | 0.006 | 0.0015 | 0.035 | 0.0038 | <0.001 | 847 | 719 | 420 | 300 |
| 3Mn | 0.20 | 1.52 | 2.98 | 0.006 | 0.0016 | 0.037 | 0.0034 | <0.001 | 797 | 689 | 363 | 220 |
| 5Mn | 0.21 | 1.50 | 4.94 | 0.005 | 0.0016 | 0.032 | 0.0020 | <0.001 | 741 | 657 | 282 | 150 |

Tensile specimens with 50 mm gauge length and 12.5 mm width parallel to the rolling direction and stretch-forming and stretch-flanging specimens with dimensions of 50 mm square were machined from the cold-rolled steel sheets. These specimens were subjected to the heat treatment shown in Figure 1b, namely, direct quenching in oil at 25 °C (DQ) and isothermal transformation (IT) at $T_{IT}$ = 100 °C to 450 °C (below $M_f$ to above $M_s$) for $t_{IT}$ = 1 × 10$^2$ s to 1 × 10$^5$ s after being austenitized at 800 °C to 900 °C for 1200 s. The IT times used are corresponding to the times for which the maximum retained austenite fraction is obtained. Hereafter, the DQ process is also dealt with as the IT process at $T_{IT}$ = 25 °C.

The microstructure of the steels was observed by field-emission scanning electron microscopy (FE-SEM; JSM-6500F, JEOL Ltd., Akishima, Tokyo, Japan), which was performed using an instrument equipped with an electron backscatter diffraction (EBSD; OIM system, TexSEM Laboratories, Inc., Prova, UT, USA) system. The EBSD analyses were conducted

in an area of 40 μm × 40 μm with a beam diameter of 1.0 μm and a beam step size of 0.1 μm operated at an acceleration voltage of 25 kV. The specimens for the FE-SEM–EBSD analysis were first ground with alumina powder and colloidal silica, and then ion thinning was carried out. The volume fraction of carbide in the specimens was measured using carbon extraction replicas and FE-SEM. The volume fractions of fine martensite/austenite constituent (MA phase) and the carbide were quantified by the line-intersecting method, as well as by the prior austenitic grain size and void density and diameter.

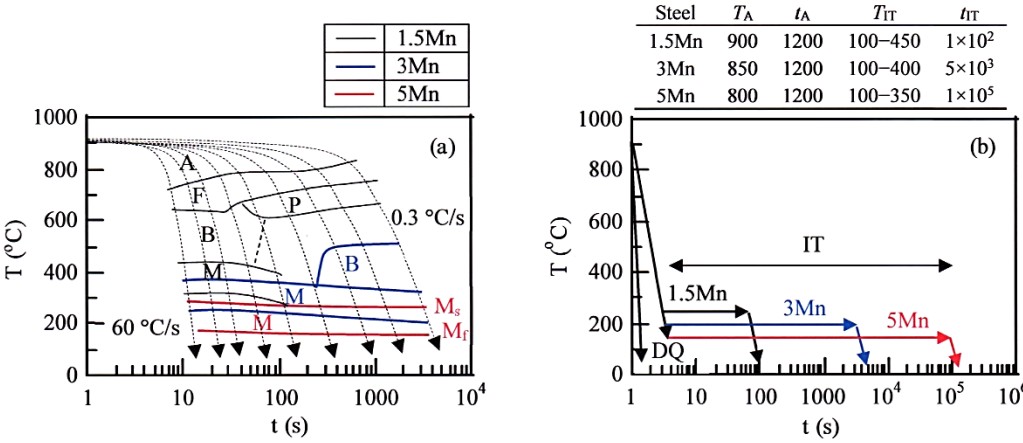

**Figure 1.** (**a**) CCT diagrams and (**b**) heat treatment diagram of the 1.5Mn, 3Mn, and 5Mn steels. "A", "F", "P", "B", and "M" in (**a**) are austenite, ferrite, pearlite, bainite, and martensite, respectively. "DQ" and "IT" represent direct quenching to room temperature and isothermal transformation, respectively. $T_A$ (°C): austenitizing temperature, $t_A$ (s): austenitizing time, $T_{IT}$ (°C): IT temperature, $t_{IT}$ (s): IT time.

The retained austenite characteristics of the steels were evaluated by an X-ray diffractometer (RINT2000, Rigaku Co., Akishima, Tokyo, Japan). The surfaces of the specimens were electropolished after being ground with emery paper (#1200). The volume fraction of the retained austenite phase ($f_\gamma$, vol.%) was quantified from the integrated intensity of the $(200)\alpha$, $(211)\alpha$, $(200)\gamma$, $(220)\gamma$, and $(311)\gamma$ peaks obtained by X-ray diffractometry using Mo-K$\alpha$ radiation [34]. The carbon concentration ($C_\gamma$, mass%) of the retained austenite was estimated from the empirical equation proposed by Dyson and Holmes [35]. To accomplish this, the lattice constant of retained austenite ($a_\gamma$, nm) was determined from the $(200)\gamma$, $(220)\gamma$, and $(311)\gamma$ peaks of the Cu-K$\alpha$ radiation. In this research, the average values of the volume fractions and carbon concentrations of retained austenite measured at three locations were adopted, as well as for other microstructural properties.

Tensile tests were carried out on a tensile-testing machine (AD-10TD, Shimadzu Co., Kyoto, Japan) at 25 °C, at a mean strain rate of $2.8 \times 10^{-3}$ s$^{-1}$ (crosshead speed: 10 mm/min). Stretch-forming tests were performed using the same testing machine used in the tensile tests to measure the maximum stretch height ($H_{max}$) at which a crack initiates. The forming temperature was 25 °C and the crosshead speed was 1 mm/min. A round punch tool with a curvature radius of 8.7 mm and a graphite lubricant were employed for the stretch-forming (Figure 2a). Hole-punching and hole-expanding tests were also conducted using the same testing machine to measure the hole-expansion ratio (HER) determined by

$$\text{HER} = (d_f - d_0)/d_0 \tag{1}$$

where $d_0$ and $d_f$ are the original hole diameter and the hole diameter upon cracking, respectively. First, a hole with a diameter of 4.76 mm was punched out at 25 °C at a punching rate of 10 mm/min, with a clearance of 10% between the die and punch using graphite lubricant (Figure 2b). Subsequently, hole expansion tests were performed using a conical punch tool with a vertical angle of 60 deg. at a crosshead rate of 1 mm/min, contacting with the roll-over section of the hole-punched specimen (Figure 2c). The hole-

expanding tests were conducted at 25 °C, using a graphite lubricant. At least two specimens were tested for each condition to obtain the tensile, stretch-forming, and stretch-flanging properties.

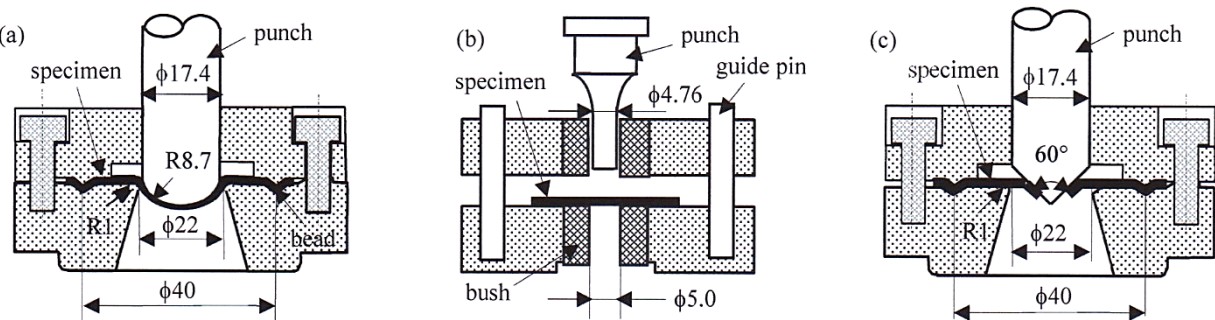

**Figure 2.** Die dimensions for (**a**) stretch-forming, (**b**) hole-punching and (**c**) hole-expanding tests.

## 3. Results

### 3.1. Microstructure

Figure 3 shows typical image quality distribution maps of the Fe-$\alpha$ (BCC) phase and FE-SEM images of carbon extraction replicas in the 1.5Mn, 3Mn, and 5Mn steels subjected to the IT process at the temperatures of $M_f$ + (30 °C to 70 °C). When the IT process was performed at the temperatures between $M_s$ and $M_f$, the microstructures of the 1.5Mn and 3Mn steels consisted of a mixture of coarse martensite and bainitic ferrite, MA phase, retained austenite, and carbide. In Figure 3a–c, the retained austenite phases look like black phases or points, because of unindexed phases or points. When the IT temperature was higher than $M_s$ and lower than $M_f$, their matrix structures changed into a single phase of bainitic ferrite or martensite, respectively. However, the matrix structure of the 5Mn steel changed into coarse martensite at all IT temperatures. In these steels, most retained austenites are mainly present in the MA phase and along the prior austenitic grain, packet, and block boundaries. The size of the coarse martensite and retained austenite phases tends to decrease with increasing Mn content. Most carbides precipitated only in coarse martensite. With increasing Mn content, both the volume fractions of the MA phase and carbide increases (see the bottom of Figure 3). The prior austenitic grain size slightly decreases with increasing Mn content, with a similar grain morphology.

Table 2 and Figure 4 show the initial volume fraction and carbon concentration of retained austenite and the strain-induced transformation factor ($k$) in the 1.5Mn, 3Mn, and 5Mn steels. The $k$ means the mechanical stability of retained austenite defined by the following equation [9],

$$k = (\log f\gamma_0 - \log f\gamma)/\varepsilon \tag{2}$$

where $f\gamma_0$ is the initial volume fraction of retained austenite and $f\gamma$ is the retained austenite fraction after plastically strained to $\varepsilon$. The volume fractions of retained austenite in the 1.5Mn and 3Mn steels increase with increasing IT temperature. On the other hand, the volume fraction of the 5Mn steel becomes peak at an IT temperature between $M_s$ and $M_f$ and drastically decreases at the IT temperatures above $M_s$. When the initial retained austenite fractions of the steels are compared at an IT temperature range between $M_s$ and $M_f$, the retained austenite fraction increases with increasing Mn content. The maximum initial carbon concentrations of the steels are obtained at the IT temperatures between $M_s$ and $M_f$. The initial carbon concentration roughly decreases with increasing Mn content. This is considered to be caused by the stabilization effect of Mn enrichment in the retained austenite [21]. Although the IT temperature dependence of $k$-value is complex, the $k$-values of the 3Mn and 5Mn steels are lower than that of the 1.5Mn steel, except for the $k$-values at $T_{IT} > 350$ °C in the 3Mn steel. The lower $k$-values of the 5Mn steel may be caused by that most of the retained austenites are finer than that of the 1.5Mn steel and surrounded by fine martensite.

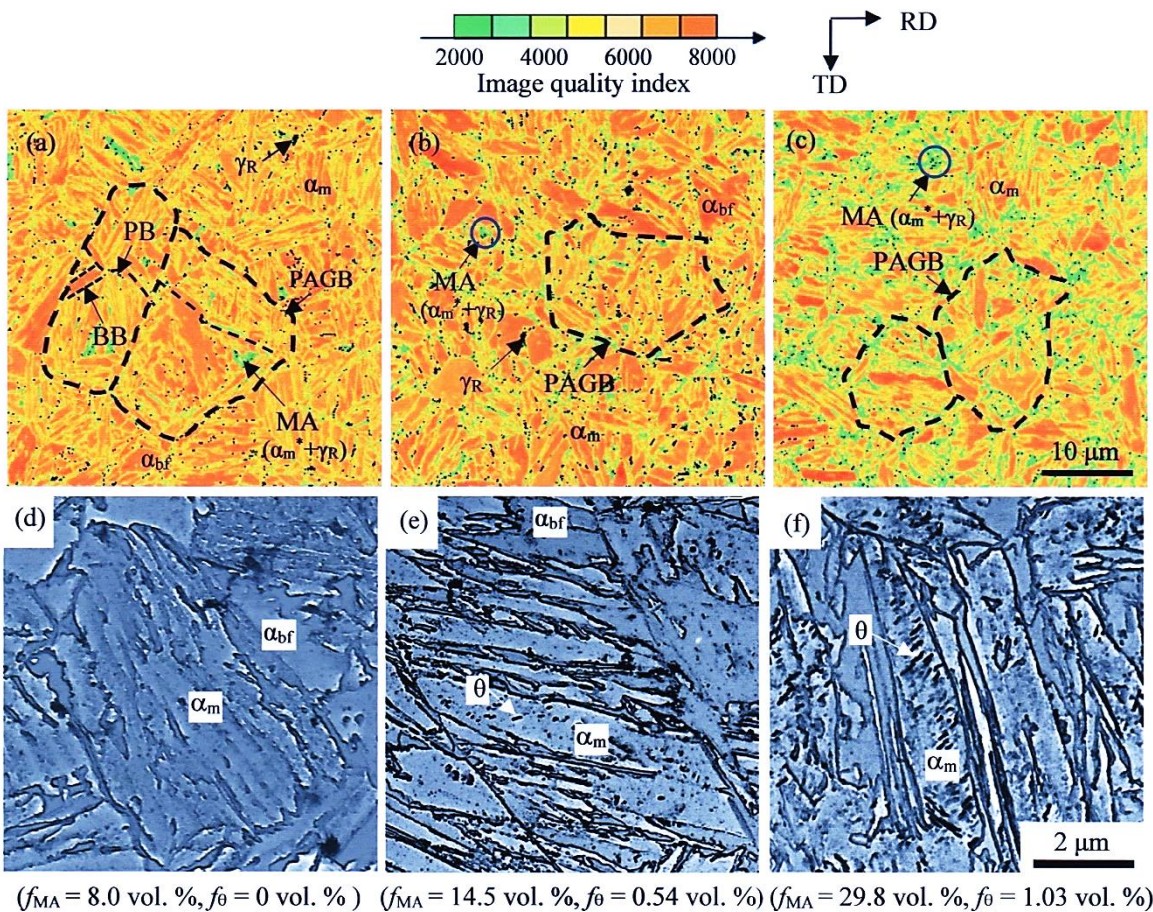

**Figure 3.** (**a**–**c**) Image-quality distribution maps of α-Fe (BCC) phase and (**d**–**f**) FE-SEM images of replicas in the 1.5Mn (**a**,**d**), 3Mn (**b**,**e**), and 5Mn (**c**,**f**) steels subjected to the IT process at $T_{IT}$ = 370 °C, 280 °C, and 180 °C, respectively. The IT temperatures correspond to $M_f$ + (30 °C to 70 °C). PAGB, PB, and BB: prior austenitic grain, packet, and block boundaries, respectively. $α_m$: coarse martensite, $α_m$*: fine martensite, $α_{bf}$: bainitic ferrite, $γ_R$: retained austenite, θ: carbide, MA: mixed phase of $α_m$* and $γ_R$. $f_{MA}$ and $f_θ$ are volume fractions of the MA phase and carbide, respectively. RD and TD represent rolling and thickness directions, respectively.

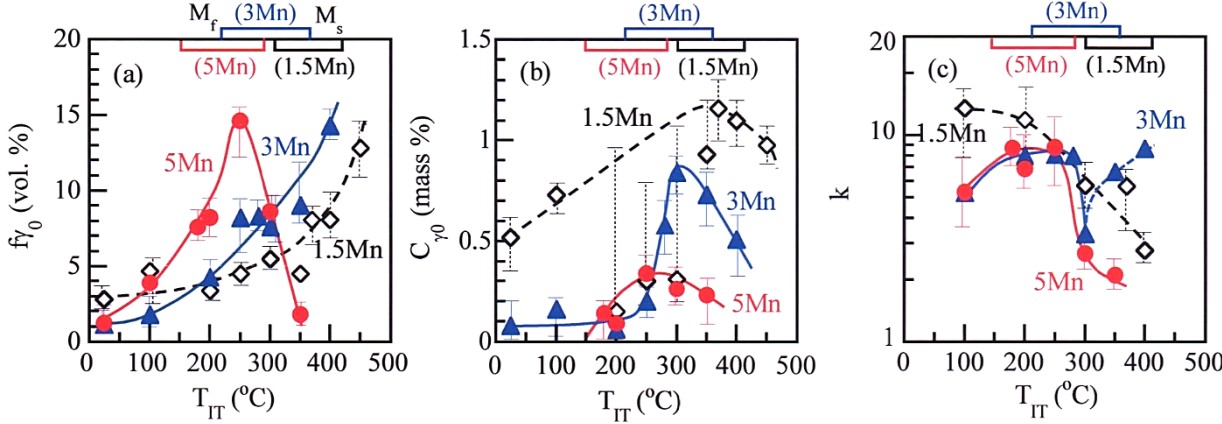

**Figure 4.** Variations in (**a**) initial volume fraction ($fγ_0$) and (**b**) carbon concentration ($Cγ_0$) of retained austenite and (**c**) strain-induced transformation factor ($k$) as a function of isothermal transformation temperature ($T_{IT}$) in the 1.5Mn (◇), 3Mn (▲), and 5Mn (●) steels.



**Table 2.** Retained austenite characteristics, tensile properties, stretch-formability, and stretch-flangeability of the 1.5Mn, 3Mn, and 5Mn steels subjected to the IT process. Gray color zones correspond to IT temperatures between $M_s$ and $M_f$ of the steels. Dotted frames correspond to the IT temperatures between $M_s$ and $M_f - 100$ °C.

| Steel | $T_{IT}$ | $f\gamma_0$ | $C\gamma_0$ | $k$ | YS | TS | UEl | TEl | RA | $H_{max}$ | TS × $H_{max}$ | HER | TS × HER |
|---|---|---|---|---|---|---|---|---|---|---|---|---|---|
| | 25 | 2.8 | 0.52 | - | 1110 | 1527 | 4.0 | 8.7 | 40.3 | 7.73 | 11.8 | 27.4 | 41.8 |
| | 100 | 4.7 | 0.73 | 13.5 | 940 | 1576 | 4.2 | 8.7 | 42.5 | 7.18 | 11.3 | 17.3 | 27.3 |
| | 200 | 3.4 | 0.15 | 11.9 | 820 | 1580 | 5.3 | 9.9 | 42.4 | 8.33 | 13.2 | 37.6 | 59.4 |
| | 250 | 4.5 | 0.30 | - | 960 | 1482 | 6.2 | 12.7 | 50.5 | 7.53 | 11.2 | 37.8 | 56.0 |
| 1.5Mn | 300 | 5.5 | 0.31 | 5.7 | 970 | 1438 | 4.3 | 10.4 | 48.0 | 8.70 | 12.5 | 42.8 | 61.5 |
| | 350 | 4.5 | 0.93 | - | 960 | 1244 | 3.7 | 10.8 | 59.9 | 8.66 | 10.8 | 44.6 | 55.6 |
| | 370 | 8.1 | 1.18 | 5.7 | 1025 | 1274 | 5.3 | 13.8 | 59.2 | 8.35 | 10.6 | 56.3 | 71.7 |
| | 400 | 8.1 | 1.16 | 2.8 | 910 | 1111 | 4.2 | 12.1 | 65.4 | 9.23 | 10.3 | 64.9 | 72.2 |
| | 450 | 12.8 | 1.10 | - | 690 | 887 | 16.2 | 24.8 | 63.7 | 9.43 | 8.4 | 59.5 | 52.8 |
| | 25 | 1.1 | 0.08 | - | 1180 | 1656 | 4.7 | 4.7 | 1.0 | 7.93 | 13.1 | 30.7 | 50.8 |
| | 100 | 1.8 | 0.16 | 5.3 | 1070 | 1603 | 6.5 | 14.1 | 54.5 | 7.92 | 12.7 | 26.1 | 41.9 |
| | 200 | 4.3 | 0.06 | 8.0 | 970 | 1546 | 8.1 | 16.0 | 51.5 | 8.20 | 12.7 | 32.9 | 50.8 |
| 3Mn | 250 | 8.2 | 0.20 | 8.0 | 940 | 1481 | 7.2 | 14.5 | 53.1 | 8.45 | 12.5 | 28.8 | 42.6 |
| | 280 | 8.3 | 0.58 | 8.0 | 1035 | 1375 | 6.9 | 14.6 | 56.5 | 8.39 | 11.5 | 30.3 | 41.7 |
| | 300 | 7.6 | 0.84 | 3.3 | 1060 | 1345 | 5.7 | 13.6 | 56.7 | 8.43 | 11.3 | 37.6 | 50.6 |
| | 350 | 9.0 | 0.73 | 6.7 | 930 | 1268 | 10.8 | 17.7 | 50.3 | 8.39 | 10.6 | 33.4 | 42.3 |
| | 400 | 14.3 | 0.51 | 8.6 | 700 | 1265 | 13.3 | 18.5 | 39.2 | 7.23 | 9.1 | 1.6 | 2.0 |
| | 25 | 1.2 | - | - | 1230 | 1907 | 5.3 | 5.3 | 5.0 | 2.40 | 4.6 | 2.7 | 5.2 |
| | 100 | 3.9 | - | 5.3 | 930 | 1757 | 7.7 | 15.4 | 54.2 | 6.26 | 11.0 | 16.7 | 29.3 |
| | 180 | 7.6 | 0.14 | 8.7 | 968 | 1633 | 7.8 | 14.6 | 46.6 | 5.74 | 9.4 | 22.9 | 37.4 |
| 5Mn | 200 | 8.2 | 0.09 | 6.9 | 970 | 1603 | 8.1 | 14.6 | 45.3 | 6.99 | 11.2 | 13.4 | 21.4 |
| | 250 | 14.6 | 0.34 | 8.7 | 870 | 1519 | 9.7 | 15.6 | 43.4 | 4.84 | 7.4 | 4.8 | 7.2 |
| | 300 | 8.6 | 0.26 | 2.7 | 1000 | 1537 | 2.0 | 2.0 | 2.0 | 1.38 | 2.1 | 1.7 | 2.6 |
| | 350 | 1.8 | 0.23 | 2.1 | 1200 | 1360 | 2.0 | 2.0 | 2.0 | 1.44 | 2.0 | 1.6 | 2.1 |

$T_{IT}$ (°C): IT temperature, $f\gamma_0$ (vol.%): initial volume fraction of retained austenite, $C\gamma_0$ (mass%): initial carbon concentration of retained austenite, $k$: strain-induced transformation factor, YS (MPa): yield stress or 0.2% offset proof stress, TS (MPa): tensile strength, UEl (%): uniform elongation, TEl (%): total elongation, RA (%): reduction of area, $H_{max}$ (mm): maximum stretch height, HER (%): hole-expansion ratio.

### 3.2. Tensile Properties

Figure 5 shows the typical engineering stress–strain curves and the instantaneous strain-hardening exponent-true strain curves of the 1.5Mn, 3Mn, and 5Mn steels subjected to the IT process at $M_f$ + (30 °C to 70 °C). Figure 6 and Table 2 show the tensile properties of these steels. The 5Mn steel has the highest tensile strength and instantaneous strain-hardening exponent (Figure 5), which result from a high concentration of Mn, the high MA fraction, and the strain-induced transformation of a large amount of retained austenite [30]. In all steels, the diffuse necking occurs at a relatively early strain, namely at half of the total fracture strain (Figure 5a), similar to conventional quenched and tempered martensitic steel. It is noteworthy that severe serration hardly appears on the flow curve of the 5Mn steel, differing from a duplex type 5Mn steel having the same chemical composition [21,36,37].

The 5Mn steel also possesses much larger uniform and total elongations (UEl and TEl) than the 1.5Mn steel at an IT temperature range between 100 °C and 250 °C, as with the 3Mn steel at an IT temperature range between 100 °C and 400 °C (Figure 6b). Note that a difference in UEl between the 1.5Mn and 5Mn steels is smaller than that in the TEl. This indicates that the 5Mn steel is characterized by larger local elongation (LEl = TEl − UEl). When the reduction of area (RA) of these steels was compared in an IT temperature range between $M_s$ and $M_f$, the RA decreases with increasing Mn content (Figure 6c). The 3Mn and 5Mn steels subjected to the DQ process (the IT process at 25 °C) showed considerably low UEl, TEl, LEl, and RA, but the IT process at 100 °C (<$M_f$) yielded high UEl, TEl, LEl, and RA, similar to those subjected to the IT process at the temperatures between $M_s$ and $M_f$.

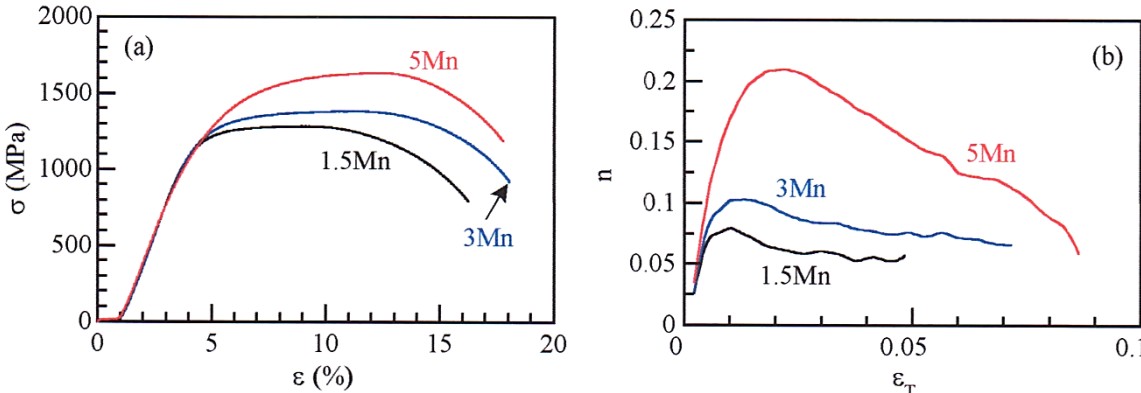

**Figure 5.** (**a**) Typical engineering stress–strain ($\sigma$-$\varepsilon$) curves and (**b**) instantaneous strain hardening exponent-true strain ($n$-$\varepsilon_T$) curves of the 1.5Mn, 3Mn, and 5Mn steels subjected to the IT process at $T_{IT}$ = 370 °C, 280 °C, and 180 °C, respectively. The IT temperatures are corresponding to $M_f$ + (30 °C to 70 °C).

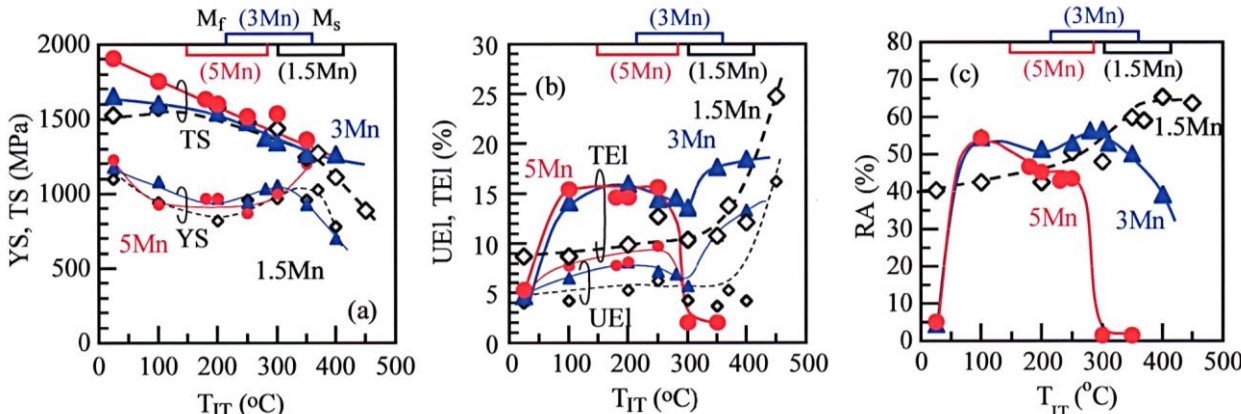

**Figure 6.** Variations in (**a**) yield stress (0.2% offset proof stress) (YS) and tensile strength (TS), (**b**) uniform (UEl) and total elongations (TEl), and (**c**) reduction of area (RA) as a function of IT temperature ($T_{IT}$) in the 1.5Mn ($\diamond$), 3Mn (▲), and 5Mn (●) steels.

Figure 7a shows the Mn-content dependences of UEl, TEl, and the products of TS and UEl and TEl (TS × UEl and TS × TEl), which are averaged values at an IT temperature range between $M_s$ and $M_f$ − 100 °C. Both products of the steels considerably increase with increasing Mn content, compared with the UEl and TEl. The TEl of the 5Mn steel is larger than that of the 1.5Mn steel and the same as that of the 3Mn steel, although UEl increases with increasing Mn content.

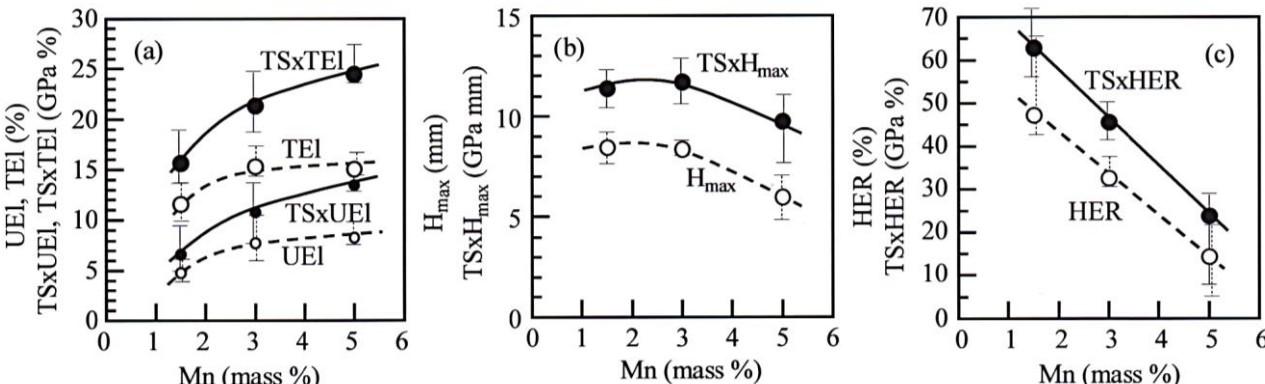

**Figure 7.** Mn content dependences of (**a**) UEl, TS×UEl, TEl, and TS × TEl and (**b**) $H_{max}$ and TS × $H_{max}$, and (**c**) HER and TS × HER which are averaged values of the steels subjected to the IT process at temperatures between $M_s$ and $M_f$ − 100 °C (corresponding to dotted frames in Table 2).

### 3.3. Stretch-Formability

Figure 8a–c shows the typical appearance of the 1.5Mn, 3Mn, and 5Mn steel samples subjected to the IT process at $M_f$ + (30 °C to 70 °C) after stretch-forming tests. The crack initiates near the punch head in all steels. Figure 9a and Table 2 show the $H_{max}$ of these steels. The $H_{max}$ of the 1.5Mn steel increases with increasing IT temperature. The IT temperature dependence of the 3Mn steel shows the same tendency as that of the 1.5Mn steel, except for $T_{IT}$ = 400 °C. The high $H_{max}$s are obtained even in the 1.5Mn and 3Mn steels with a tensile strength above 1500 MPa. The $H_{max}$s of the 5Mn steel are lower than those of the 1.5Mn and 3Mn steels. Moreover, the IT temperature dependence of $H_{max}$ is different from those of the 1.5Mn and 3Mn steels. That is, the $H_{max}$ achieves the peak values at the IT temperatures just above and below $M_f$. When subjected to the DQ process, only the 5Mn steel possesses a much low $H_{max}$.

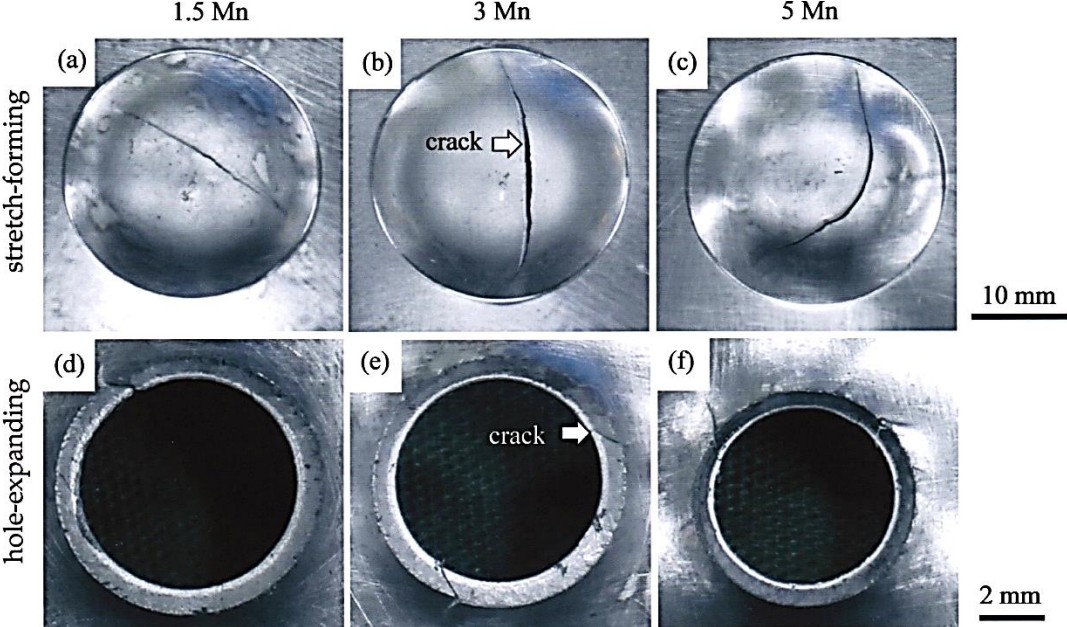

**Figure 8.** Typical samples of the 1.5Mn, 3Mn, and 5Mn steels respectively subjected to the IT process at 370 °C, 280 °C, and 180 °C after stretch-forming (**a**−**c**) and hole-expanding tests (**d**−**f**). The IT temperatures are corresponding to $M_f$ + (30 °C to 70 °C).

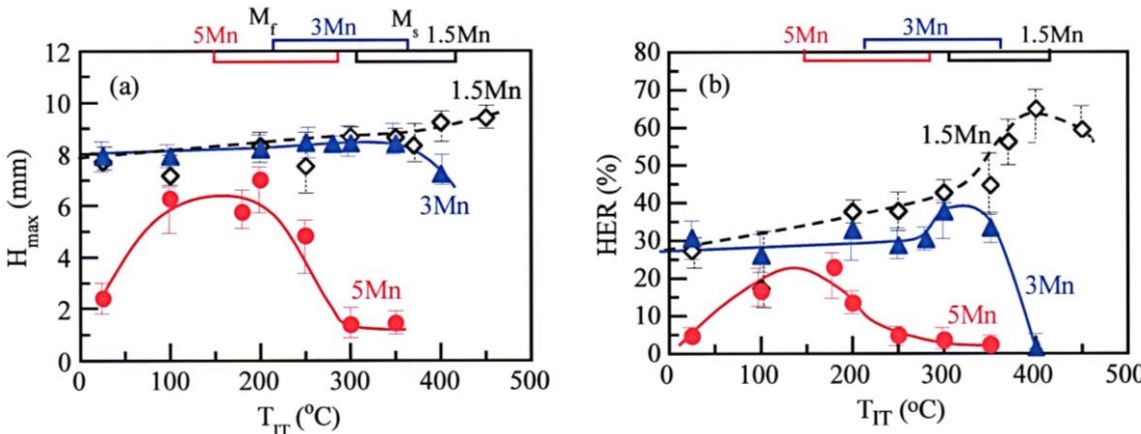

**Figure 9.** Variations in (**a**) maximum stretch height ($H_{max}$) and (**b**) hole-expansion ratio (HER) as a function of isothermal transformation temperature ($T_{IT}$) in the 1.5Mn ($\diamond$), 3Mn ($\blacktriangle$), and 5Mn ($\bullet$) steels.

Figure 7b shows the average values of the $H_{max}$ and the products of TS and $H_{max}$ (TS × $H_{max}$) of the 1.5Mn, 3Mn, and 5Mn steels subjected to the IT process at the temperatures between $M_s$ and $M_f$ − 100 °C. The Mn content dependence of TS × $H_{max}$ is relatively small, although the peak value is obtained in the 3Mn steel.

### 3.4. Stretch-Flangeability

Figure 10 shows the typical shear stress-displacement curves on hole-punching in the 1.5Mn, 3Mn, and 5Mn steels subjected to the IT process at $M_f$ + (30 °C to 70 °C). The shear stress increases with increasing Mn content, although it decreases after peak shear stress in the 3Mn and 5Mn steels. The final punching displacement of the 1.5Mn steel is slightly larger than those of the 3Mn and 5Mn steels.

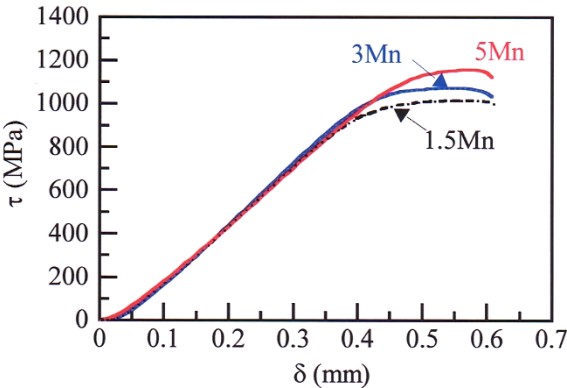

**Figure 10.** Typical shear stress-displacement ($\tau$-$\delta$) curves on hole-punching in 1.5Mn, 3Mn, and 5Mn steels subjected to the IT process at 370 °C, 280 °C, and 180 °C, respectively. The IT temperatures are corresponding to $M_f$ + (30 °C to 70 °C).

Figure 11 shows SEM images of the break section after hole-punching in the 1.5Mn, 3Mn, and 5Mn steels subjected to the IT process at $M_f$ + (30 °C to 70 °C). A large number of voids are formed in the punching surface layer of these steels, accompanied by significant plastic flow. The shear section length decreases with increasing Mn content (see the bottom of Figure 11). When the void properties are measured in a region from surface to 50 μm depth, the average void diameter increases with increasing Mn content, although the void density decreases with increasing Mn content.

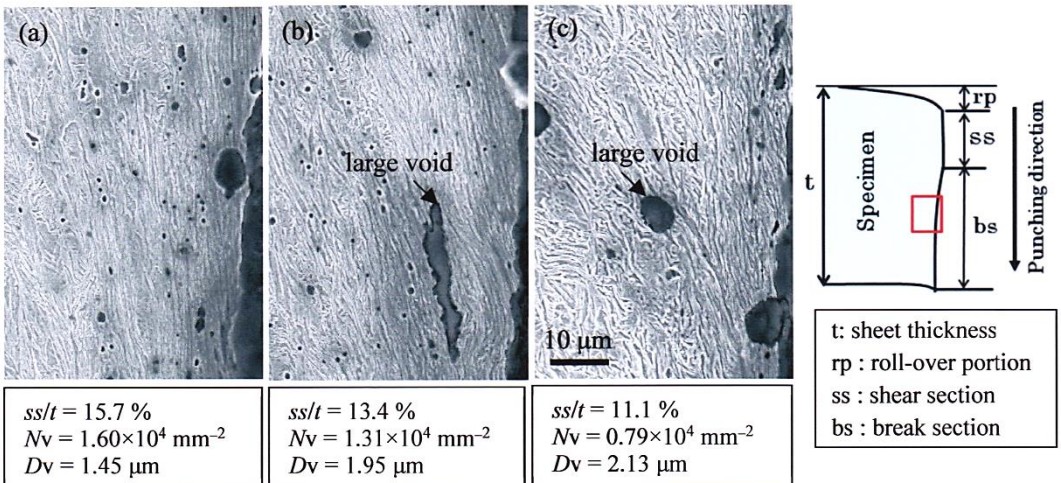

**Figure 11.** SEM images of break section after hole punching of (**a**) the 1.5Mn, (**b**) 3Mn, and (**c**) 5Mn steels subjected to the IT process at $T_{IT}$ = 370 °C, 280 °C, and 180 °C, respectively. The IT temperatures are corresponding to $M_f$ + (30 °C to 70 °C). $ss/t$: a ratio of shear section length ($ss$) to thickness ($t$), $N_v$: void density, $D_v$: average void diameter.

Figure 8d–f shows the typical appearance of the 1.5Mn, 3Mn, and 5Mn steel samples subjected to the IT process at $M_f$ + (30 °C to 70 °C) after hole-expanding tests. A few cracks initiate at the punched surface without necking in all steels. Figure 9b and Table 2 show the HER of these steels. The IT temperature dependences of HER of the 1.5Mn, 3Mn, and 5Mn steels resemble those of the $H_{max}$ (Figure 9b). However, these IT temperature dependences appear more prominent than those of the $H_{max}$. In the same way as the UEl, TEl, and RA of Figure 6b,c, the 5Mn steel subjected to the DQ process possesses a much low HER.

Figure 7c shows the average values of the HER and the products of TS and HER (TS $\times$ HER) of the 1.5Mn, 3Mn, and 5Mn steels subjected to the IT process at the temperatures between $M_s$ and $M_f$ − 100 °C. The Mn-content dependence of the TS $\times$ HER drastically decreases with increasing Mn content, differing from the TS $\times$ $H_{max}$. The value of the 5Mn steel is half that of the 1.5Mn steel. This indicates that the 5Mn steel is disfavorable for cold hole expansion despite a good tensile ductility.

## 4. Discussion

In general, stress states of sheet forming can be classified into four modes [38], namely

- deep drawing [stretch (tension) and shrink (compression)],
- stretch-forming [equi-biaxial stretch (tension)],
- stretch-flanging [stretch (tension)] and
- bending [stretch (tension) and bending].

In a case of stretch-flanging, hole-punching by shearing is conducted before hole-expanding. Moreover, these formabilities are influenced by the following microstructural properties in the third generation AHSSs [27–29].

(i)  volume fraction and mechanical stability of retained austenite which control the strain-induced martensite transformation hardening and plastic relaxation of the localized stress concentration (or TRIP effect),

(ii)  MA phase properties such as volume fraction, hardness, etc. which influences the internal stress hardening and void/crack-initiation and -growth behavior at the matrix/MA phase. For the M–MMn steel, the following hardening takes part in the deformation [21],

(iii)  Mn concentration or content which contributes to the solid solution hardening and influences the sizes of prior austenitic grain, coarse martensite, and retained austenite, as well as the prior austenitic grain boundary strength.

In the following, the stretch-formability and stretch-flangeability of the 1.5Mn, 3Mn, and 5Mn steels are related to the microstructural properties (i) to (iii), as well as the tensile ductility.

### 4.1. Relationship between Tensile Ductility and Microstructural Properties

The UEl, TS $\times$ UEl, TEl, and TS $\times$ TEl of the present steels increased with increasing Mn content when they were subjected to the IT process at temperatures between $M_s$ and $M_f$ − 100 °C. The TS $\times$ UEl and TS $\times$ TEls were constant or decreased with increasing retained austenite fraction (Figure 12a) and increased with increasing $k$-values, except for the TS $\times$ TEl of the 5Mn steel (Figure 12b). Such results have been also found in 0.2%C-(1.0−2.5)%Si-(1.0−2.0)%Mn TPF steels tested at room temperature [39]. Sugimoto et al. [39,40] found that the TS $\times$ TEl at optimum warm testing temperatures at which the mechanical stability of the retained austenite becomes the maximum linearly increased with increasing initial volume fraction of retained austenite in the TPF steels. So, the results of Figure 12 are considered to be mainly caused by the low mechanical stability of the retained austenite, not the retained austenite fraction.

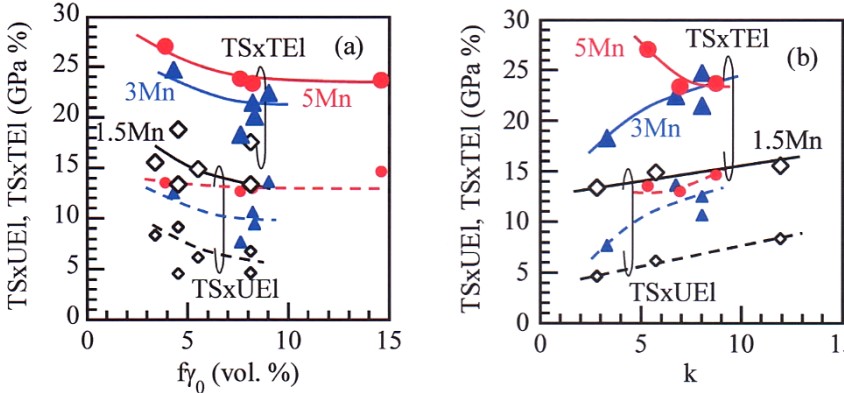

**Figure 12.** Relationships between TS × TEl and TS × UEl and (**a**) initial volume fraction ($f\gamma_0$) and (**b**) strain-induced transformation factor ($k$) of retained austenite in the 1.5Mn (◇), 3Mn (▲), and 5Mn (●) steels. The IT temperatures are between $M_s$ and $M_f - 100\,°C$.

According to Kobayashi et al. [27], Sugimoto et al. [28], and Pham et al. [29] found that a given volume fraction of MA phase fraction (10–15 vol.%) improves the TS × TEl in 0.2%C-1.5%Si-1.5%Mn-1.0%Cr-0.05%Nb TM steel [28]. In the present research, the MA phase fraction increased with increasing Mn content (Figures 3 and 13b). Therefore, the high TS × TEl of the 5Mn steel is considered to be mainly associated with the large strain-hardening behavior (Figure 5b) resulting from the (i), (ii), and (iii), accompanied with the plastic relaxation of the retained austenite which lowers the localized stress concentration at the MA phase/matrix interface and resultantly causes the difficult void/crack-initiation, even high MA phase fraction. The degree of contribution of the (i) to (iii) to the tensile ductility is summarized in Table 3.

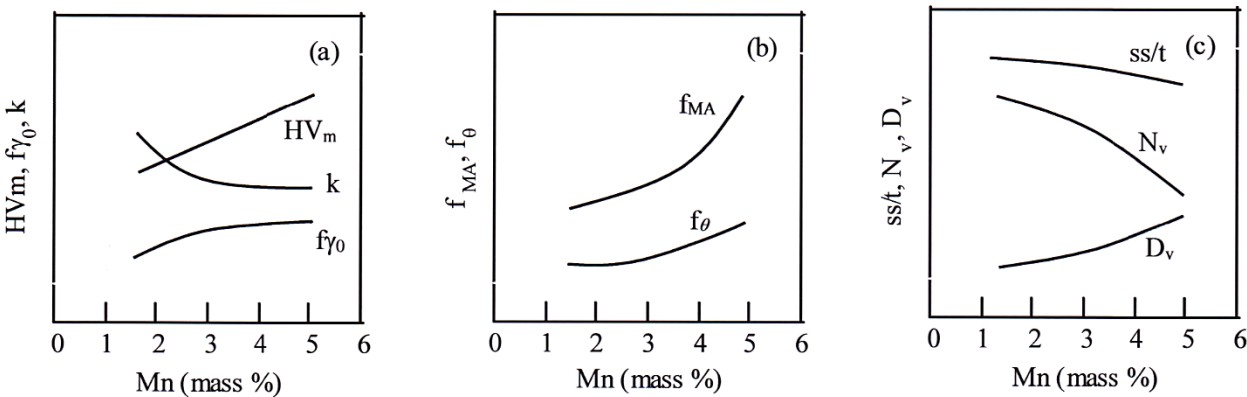

**Figure 13.** Schematic Mn content dependences of (**a**) Vickers hardness of matrix ($HV_m$) and initial volume fraction ($f\gamma_0$) and strain-induced transformation factor ($k$) of retained austenite, (**b**) volume fractions of MA phase ($f_{MA}$) and carbide ($f_\theta$), and (**c**) a ratio of shear section length to sheet thickness ($ss/t$), void density ($N_v$) and average void diameter ($D_v$) in the steels subjected to the IT process at temperatures of $M_f + (30\,°C$ to $70\,°C)$.

**Table 3.** Degree of the contribution of microstructural property to formability in the 5Mn steel.

| Microstructural Property | Tensile Ductility | Stretch-Formability | Stretch-Flangeability | |
|---|---|---|---|---|
| | | | Hole-Punching | Hole-Expanding |
| (i) retained austenite | significant increase | increase | increase | slight increase |
| (ii) MA phase | increase | significant decrease | significant decrease | significant decrease |
| (iii) Mn concentration | increase | increase | unknown | unknown |
| Total of (i) to (iii) | significant increase | decrease | significant decrease | |

In Figure 6b, the UEl and TEl of the 5Mn steel subjected to the DQ process (the IT process at $T_{IT}$ = 25 °C) were significantly small compared to those subjected to the IT process at the temperatures between $M_s$ and $M_f$ − 100 °C. This may be mainly caused by the lower volume fraction and lower carbon concentration (mechanical stability) of retained austenite (Figure 4a,b).

### 4.2. Relationship between Stretch-Formability and Microstructural Properties

As well known, the stretch-formability of the third-generation AHSSs is also controlled by the above (i) to (iii). However, under an equi-biaxial stress state, the strain-induced transformation of the retained austenite and void/crack-initiation at the MA phase/matrix interface are promoted, compared with those under a uniaxial tension-stress state [40]. Resultantly, stretch-formability is significantly influenced by strain-induced transformation and the void/crack-formation behaviors, compared with tensile ductility.

In Figure 7b, the 5Mn steel showed lower $H_{max}$ and TS × $H_{max}$ than the 1.5Mn and 3Mn steels, despite possessing higher tensile ductility. As shown in Figure 14a, the TS × $H_{max}$ of the 1.5Mn and 3Mn steels shows a positive linear relationship with the TS × UEl, although the TS × $H_{max}$ of the 5Mn steel was lower than those of the 1.5Mn and 3Mn steels. As shown in Figure 14b,c, the TS × $H_{max}$ of the present steels decreases with increasing retained austenite fraction and increases with increasing *k*-value, in the same way as the TS × UEl and TS × TEl (Figure 12). The volume fraction and mechanical stability of retained austenite dependences of the TS × $H_{max}$ differ from that reported by Sugimoto et al. [39], while the retained austenite with high mechanical stability considerably increased the $H_{max}$ and TS × $H_{max}$ in 0.2%C-(1.0−2.5)%Si-(1.0−2.0)%Mn TPF steels. Therefore, these dependencies may be caused by the low mechanical stability. In this case, the retained austenite contributes to the increasing of the TS × $H_{max}$. The 5Mn steel contained a larger amount of harder MA phase. Therefore, it is considered that the MA phase promotes the easy void/crack-initiation at the MA phase/matrix interface resulting from the equi-biaxial stress state and mainly decreases the stretch-formability. The solid-solution hardening of Mn maybe also contribute to increasing its stretch-formability. The degree of contribution of the (i) to (iii) on the stretch-formability is summarized in Table 3. It is considered that the carbides in the 5Mn steel (Figure 3f) hardly influence the $H_{max}$ and TS × $H_{max}$ because the amount is small and the crack initiation site is MA phase/matrix interface.

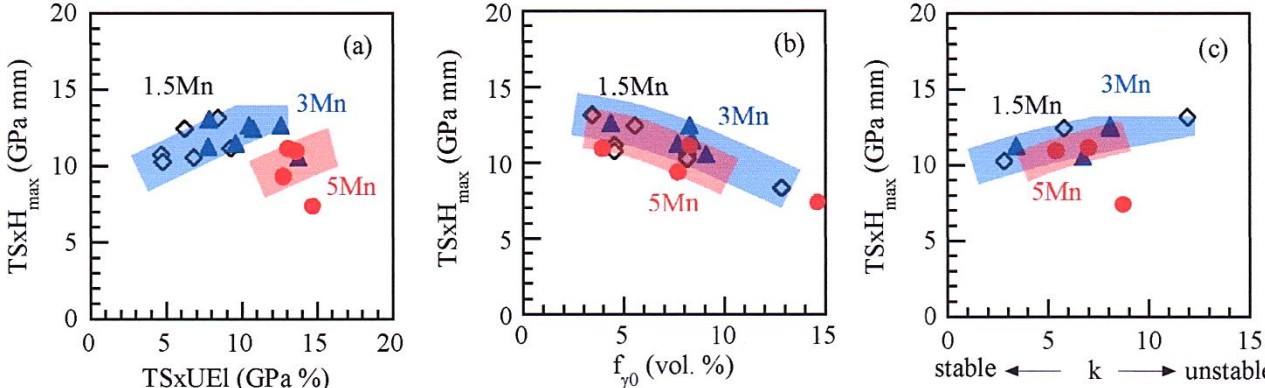

**Figure 14.** Relationships between TS × $H_{max}$ and (**a**) TS × UEl, (**b**) initial retained austenite fraction ($f\gamma_0$), and (**c**) strain-induced transformation factor (*k*) in the 1.5Mn (◇), 3Mn (▲), and 5Mn (•) steels. IT temperatures are between $M_s$ and $M_f$ − 100 °C.

### 4.3. Relationship between Stretch-Flangeability and Microstructural Properties

According to Sugimoto et al. [7,8,26–28], the stretch-flangeability of the third generation AHSS sample can be mainly evaluated by

(1) surface layer damage suffered on hole-punching (severe plastic flow, strain hardening behavior, and void/crack-initiation damage) and

(2) ductility and crack/void-growth and -connection behaviors on hole-expanding.

The (1) and (2) are controlled by the various metallurgical characteristics, such as hardness of the matrix structure, MA phase properties, a hardness ratio of the second phase to the matrix, micro-scale uniformity, retained austenite characteristics, carbide properties, etc. In the third-generation AHSS, the (1) plays a dominating role in the stretch-flangeability. When the 1.5Mn, 3Mn, and 5Mn steels were subjected to the IT process at the temperatures between $M_s$ and $M_f - 100\ °C$, the averaged values of HER and TS × HER linearly decreased with increasing Mn content (Figure 7c). In the following, the relationship between the stretch-flangeability and the metallurgical characteristics is discussed.

First, let us discuss the effect of hole-punching damage on stretch-flangeability. When the present steels were subjected to the IT process at the temperatures between $M_s$ and $M_f - 100\ °C$, the 3Mn and 5Mn steels possessed a larger amount of stable retained austenite than the 1.5Mn steel (Figures 4 and 13a). Also, the microstructure of the 5Mn steel can be characterized by higher matrix hardness (lower image quality in Figure 3c) and higher volume fractions of MA phase and carbide than those of the 1.5Mn steel (Figures 3 and 13b). The hardness ratio of the MA phase to the coarse martensite matrix structure is estimated to be relatively high because the image quality of the MA phase is significantly low (Figure 3c). Such microstructural characteristics may promote the void/crack-initiation on hole-punching. The largest voids were formed in a hole-punched surface layer of the 5Mn steel, although the number of voids was minimum (Figures 11 and 13c). Moreover, the shear section length was the smallest. Thus, the high MA phase fraction and the high hardness ratio may result in short shear-section length and large void sizes on hole-punching in the 5Mn steel. As shown in Figure 10, the final punching displacement of the 5Mn steel was slightly smaller than that of 1.5Mn steel, although the punching shear stress was higher than that of 1.5Mn steel. This means that the TRIP effect of the retained austenite contributes to the reduction of the punching damage (see Table 3).

Next, we discuss the effect of hole-expanding on the stretch-flangeability of the 5Mn steel. In general, hole-expanding of the third generation AHSSs is mainly controlled by void/crack growth and connection [27–29]. As most retained austenites near the punched-hole surface transform to martensite on hole-punching, the untransformed retained austenite is difficult to contribute to the suppression of the void/crack-growth and -connection on hole-expanding. Therefore, it is considered that the hole-surface damage on punching resulting from the MA phase (referring to the (ii)) may control its stretch-flangeability, with a slight contribution from the retained austenite (see Table 3).

The TS × HER increased with increasing TS × RA in the present steels (Figure 15a). In addition, the TS × HER increases with increasing initial volume fraction of retained austenite (the line I in Figure 15b) and decreasing *k*-value (or increasing mechanical stability of the retained austenite (Figure 15c). This indicates that the retained austenite plays a positive role in its stretch-flangeability by suppressing the void-initiation on hole-punching, because it plays a small role in hole-expanding. Therefore, the (ii) preferentially decreases the HER by the large hole-punching damage and easy void-growth and -connection on hole-expanding in the 5Mn steel (Table 3), despite the positive contribution from a large amount of retained austenite. As carbide fraction was relatively small, the effect of carbide on stretch-flangeability is considered to be negligible. Unfortunately, the effect of the (iii) on stretch-flangeability is not clear at present.

At the present stage, the discussion about the relationship between stretch-flangeability and the microstructural properties is not perfect, in the same way as the stretch-formability. Further detailed investigation of the relationship is expected in the future.

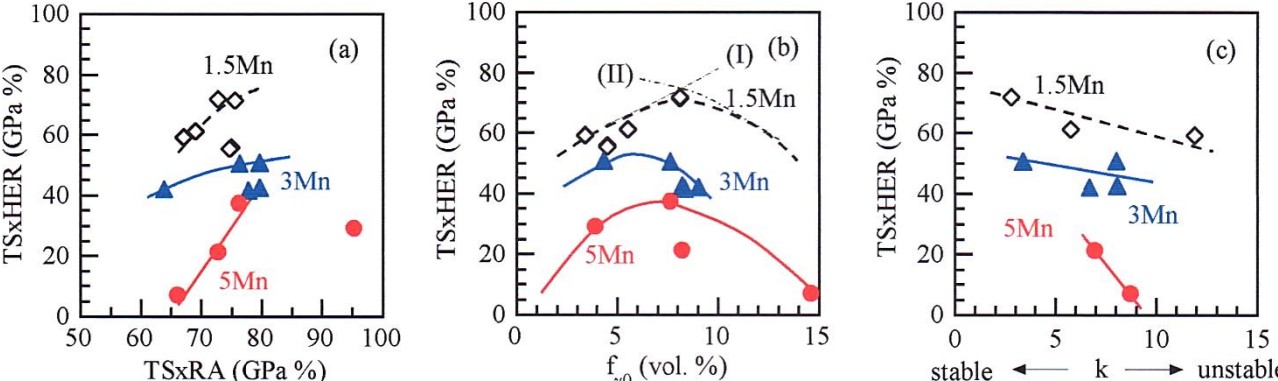

**Figure 15.** Relationships between TS × HER and (**a**) TS × RA, (**b**) initial retained austenite fraction ($f\gamma_0$), and (**c**) strain-induced transformation factor ($k$) in the 1.5Mn (◇), 3Mn (▲), and 5Mn (●) steels. The IT temperatures are between $M_s$ and $M_f − 100\,°C$. Dashed lines I and II in (**b**) represent positive and negative $f\gamma_0$ dependences of TS × HER, respectively.

*4.4. Further Improvement of Stretch-Formability and Stretch-Flangaebility*

Finally, we discuss how to improve the $H_{max}$ and HER of the 5Mn steel. According to Sugimoto et al. [35], warm hole-punching and/or hole-expanding at 200 °C significantly increase the TS × HER of duplex type 3Mn and 5Mn steels with reverted austenites of 24 and 40 vol.%, respectively. Although the present martensite-type 3Mn and 5Mn steels possess relatively low retained-austenite fractions, it is expected that warm hole-punching and/or hole-expanding improve the stretch-flangeability because the warm working or heating increases the mechanical stability of the retained austenite and softens the matrix and MA phase. Resultantly, the void/crack-initiation, -growth, and -connection are suppressed and shear-section length is increased. In the same way, warm forming is expected to increase the stretch-formability.

In this work, the DQ process considerably decreased the $H_{max}$ and HER of the 5Mn steel. According to Kobayashi et al. [27], partitioning (or tempering) after the DQ process promotes carbon enrichment in the retained austenite and softening of the coarse and fine martensites, with a slight increase in carbide fraction. Thus, this partitioning is also considered to enhance the $H_{max}$ and HER of the 5Mn steel subjected to the DQ process. According to Zheng et al. [41], the easiest way to improve the $H_{max}$ and HER of the medium Mn steel is lowering the carbon content.

## 5. Conclusions

The cold stretch-formability and stretch-flangeability of the martensite type 5Mn steel subjected to the IT process at the temperatures from above $M_s$ to below $M_f$ were investigated and were related to the microstructural properties, as well as tensile ductility. The main results can be summarized as follows:

(1) The highest UEl and TEl were achieved in the 5Mn steel subjected to the IT process at temperatures between $M_s$ and $M_f − 100\,°C$. The TS × UEl and TS × TEl were two and one-half times those of the 1.5Mn steel (equivalent to the TM steel), respectively. This was mainly associated with the TRIP effect of a large amount of retained austenite, accompanied by high strain-hardening by a large amount of MA phase and high Mn content.

(2) High $H_{max}$ was obtained in the 5Mn steel subjected to the IT process at temperatures between $M_s$ and $M_f − 100\,°C$. However, the TS × $H_{max}$ was slightly lower than those of the 1.5Mn and 3Mn steels. In this case, a large amount of metastable austenite and high Mn content contributed to increasing stretch-formability. However, the presence of a large amount of MA phase significantly reduced stretch-formability through facilitating void/crack initiation and growth.

(3) In the 5Mn steel, the IT temperature dependence of the HER resembled that of the $H_{max}$. However, the optimum TS × HER obtained by the IT process at the

temperatures between $M_s$ and $M_f - 100\ ^\circ\text{C}$ considerably decreased compared to those of the 1.5Mn and 3Mn steels. The decreased stretch-flangeability of the 5Mn steel was mainly associated with large hole-punching surface damage, such as short shear length and the presence of large voids, mainly caused by the high volume fraction and hardness of the MA phase. The retained austenite contributed to lowering the hole-punching damage. The effect of Mn content on the stretch-flangeability was not clear at present.

**Author Contributions:** The first draft of the paper was written by K.-i.S. and the final editing was done by K.-i.S., H.T. and J.K. All authors have read and agreed to the published version of the manuscript.

**Funding:** This research received no external funding.

**Institutional Review Board Statement:** Not applicable.

**Informed Consent Statement:** Not applicable.

**Data Availability Statement:** Not applicable.

**Acknowledgments:** We thank Tomohiko Hojo from Tohoku University for the kind discussion.

**Conflicts of Interest:** The authors declare no conflict of interest.

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
