# Peer review of "Cold Formabilities of Martensite-Type Medium Mn Steel"

_metals, doi:10.3390/met11091371_

Round 1

Reviewer 1 Report

This is a research on cold formabilities of martensite-type medium Mn steel,   including many tests of mechanical properties and cold formabilities. The matter is interesting, but there are some unclear parts which are listed below:

  1. The description of the total elongation of 5Mn steel in the abstract is inappropriate. The total elongation of 5Mn steel is not higher than that of the other two experimental steels.
  2. The last sentence of the abstract lacks support, because there are many factors that affect the formability, such as the hardness difference between phases, grain size and grain morphology, etc., which are not considered in the article.
  3. The analysis of formability lacks the support of microstructure characterization in the main part of the manuscript.
  4. How to determine the volume fractions of MA and θ in the paper? Please give the description of quantitative method. Please mark the coarse lath-martensite, fine lath/twin martensite/austenite constituents (MA phase), and retained austenite in the figure 3. Please provide the data of characterization of microstructure, XRD diffraction pattern, which is very important.
  5. The description of Figure 7 in this paper is not accurate as TEl does not increase with the increase of Mn content.
  6. Degree of the contribution of microstructural property to formability in Table 3 is a subjective judgment and lacks scientific quantitative characterization.

Author Response

Author’s Reply to Reviewer 1

Very Thanks for your kind review. Following your comments, the authors reply as follows. The revised word and sentences are highlighted by the red ribbons or bands in the revised manuscript.

This is research on cold formabilities of martensite-type medium Mn steel, including many tests of mechanical properties and cold formabilities. The matter is interesting, but there are some unclear parts which are listed below:

  1. The description of the total elongation of 5Mn steel in the abstract is inappropriate. The total elongation of 5Mn steel is not higher than that of the other two experimental steels.

[Reply] As shown in Figure 7, a total elongation of the 5Mn steel is larger than that of 1.5Mn steel and the same as that of 3Mn steel. So, “or the same” was added as follows.

Both formabilities of the steel decreased compared to those of 0.2%C-1.5%Si-1.5Mn and -3Mn steels, despite a larger or the same total elongation, especially in the stretch-flangeability.

  1. The last sentence of the abstract lacks support, because there are many factors that affect the formability, such as the hardness difference between phases, grain size and grain morphology,, which are not considered in the article.

[Reply] In this research, the formability of the M-Mn steel is assumed to be mainly controlled by (i) the retained austenite characteristics, (ii) MA phase properties and (iii) Mn concentration or content. Because the effect of the hardness difference between phases is included in the (ii) and (iii), it was excluded from the abstract and conclusions. As the effect of the prior austenitic grain size and morphology is included in the (iii) and relatively small, it was also excluded from the abstract and conclusions.

The description of the prior austenitic grain properties was added to line 210.

Because the effect of Mn content was not described in the abstract, the following sentence was added at the end of the abstract.

High Mn content contributed to increasing the stretch-formability.

In addition, the description of the effect of the Mn content on the formabilities was added in conclusions (2) and (3).

  1. The analysis of formability lacks the support of microstructure characterization in the main part of the manuscript.

[Reply] We also feel so. It is very difficult to understand the relationship between the formabilities and the microstructural properties. At the present stage, the discussion about the relationship is not perfect. We think to need further investigation of the relationship in the future. To state this, the authors add the following sentence to line 501.

At the present stage, the discussion about the relationship between the stretch-flangeability and the microstructural properties is not perfect, in the same way as the stretch-formability. Further detailed investigation of the relationship is expected in the future.

  1. How to determine the volume fractions of MA and θ in the paper? Please give the description of quantitative method.

[Reply] The volume fractions of MA phase and carbide were measured by the line intersecting method, as well as the prior austenitic grain size and void density and diameter. This was added to line 108.

  1. Please mark the coarse lath-martensite, fine lath/twin martensite/austenite constituents (MA phase), and retained austenite in the figure 3.

[Reply] It is difficult to classify the lath and twin martensite in Figure 3. So, the authors deleted “the lath-twin”. Hereafter, we call the matrix structure “coarse martensite (αm)” and martensite in MA phase “fine martensite” (αm*). The coarse martensite is already marked in Figure 3, but it is difficult to mark the fine martensite because of its very fine size.

  1. Please provide the data of characterization of microstructure, XRD diffraction pattern, which is very important.

[Reply] XRD patterns are important to understand the retained austenite characteristics. However, the values of the volume fraction and mechanical stability of the retained austenite are more important. Because the aim of the present paper is the formabilities of M-MMn steel, we would like to show only these values. We hope to show the XRD profiles in another paper.

  1. The description of Figure 7 in this paper is not accurate as TEl does not increase with the increase of Mn content.

[Reply] The following sentence was added to line 270.

The TEl of the 5Mn steel is larger than that of the 1.5Mn steel and the same as that of the 3Mn steel, although the UEl increases with increasing Mn content.

  1. Degree of the contribution of microstructural property to formability in Table 3 is a subjective judgment and lacks scientific quantitative characterization.

[Reply] The opinion by the reviewer is correct. We think that the contribution of microstructural property to formability should be quantitatively evaluated in the future. At the present stage, discussion using Table 3 is the limit. As replied at Comment 3, the following sentence was added to line 501.

Further detailed investigation of the relationship is expected in the future.

  1. Others

Ref. 41 was added. Please check the sentence of line 637 and References.

That’s all.

Reviewer 2 Report

The paper investigated cold stretch-formability and stretch-flangeability of 1.5, 3 and 5 mass % Mn steel. The experiments of the study are adequate and the analysis/discussion are reasonable. This manuscript can be accepted for publication if the following points are taken into consideration.

  1. Page 2 line 67, “The CCT” instead of “CCT”. Please check again carefully the use of articles throughout the manuscript.
  2. In Fig.4, Fig. 6, Fig. 7, Fig.9, Fig. 12 and Fig.14, using different symbols rather than different colors in the graphic annotation is better.
  3. Considerable improvements of Fig.13 in the manuscript can be made by image editing software.
  4. The method of determination of void density and void diameter is missing.
  5. Re-work the English language in the conclusions part to make the conclusions more logical and clear.

Author Response

Author’s Reply to Reviewer 2

Thanks for your kind review. Following your comments, the authors reply as follows. The revised word and sentences are highlighted by the blue ribbons or bands in the revised manuscript.

The paper investigated cold stretch-formability and stretch-flangeability of 1.5, 3 and 5 mass % Mn steel. The experiments of the study are adequate and the analysis/discussion are reasonable. This manuscript can be accepted for publication if the following points are taken into consideration.

  1. Page 2 line 67, “The CCT” instead of “CCT”. Please check again carefully the use of articles throughout the manuscript.

[Reply] Author revised it. We made careful check again.

  1. In Fig.4, Fig. 6, Fig. 7, Fig.9, Fig. 12 and Fig.14, using different symbols rather than different colors in the graphic annotation is better.

[Reply] In Figs.4, 6, 9, 12, 14 and 15, the authors changed the symbols ot the 1.5Mn and 3Mn steels. Also, the captions of these figures were revised.

  1. Considerable improvements of Fig.13 in the manuscript can be made by image editing software.

[Reply] Thanks for your comment. Unfortunately, we do not know “the image editing software”. I am sorry that I cannot reply to your comment. I hope to keep this figure. However, please teach me how to revise the figure if the revision is needed.

  1. The method of determination of void density and void diameter is missing.

[Reply] The void density and void diameter were measured in a region from surface to 50 μm depth by the line intersecting method. The method was added to lines 108 and 386.

  1. Re-work the English language in the conclusions part to make the conclusions more logical and clear.

[Reply] The conclusions were re-checked and modified as much as possible.

  1. Others

Ref. 41 was added. Please check the sentence of line 637 and References.

That’s all

Reviewer 3 Report

The paper is well structured and discusses scientifically on formability of low and medium manganese steel grades under different thermal cycles. However, it needs to be improved to be accepted for publication.  Some comments and questions which can improve the quality of the work can be found below:

Introduction can be improved to show that what is the current research gap on the formability of Med-Mn steels.

What was the step size for the EBSD measurement? Some of the areas pointed out as the retained austenite seem to be unindexed points like Figure 3(C).

This sentence ‘’ fMA and fθ are volume fractions of the MA phase and carbide ’’ should be removed from the Figure 3 caption as it is not related to this figure.

One of the critical steps to have high-performance Med-Mn steels is the intercritical annealing prior to the final treatment, why it has not been considered in this work?

Considering the tensile properties, the mechanical properties are very similar to the maraging TRIP steels and far from the Med-Mn steels

The effect of grain size (both bcc and fcc phase) on the achieved mechanical properties can add more clarification on the results obtained from the different processing routes and alloys which have been used in this work.

Author Response

Author’s Reply to Reviewer 3

Very thanks for your kind review. Following your comments, the authors reply as follows. The revised word and sentences are highlighted by the green ribbons or bands in the revised manuscript.

The paper is well structured and discusses scientifically on formability of low and medium manganese steel grades under different thermal cycles. However, it needs to be improved to be accepted for publication.  Some comments and questions which can improve the quality of the work can be found below:

  1. Introduction can be improved to show that what is the current research gap on the formability of Med-Mn steels.

[Reply] The sentence filling the current research gap was added to line 51.  

If the M-Mn steel achieves higher cold formability than the TM and D-MMn steels, significant weight reduction of the automobiles is expected. Many researchers reported the formability of the D-MMn steel because the steel achieves superior formability [22]. However, the cold formabilities of the M-MMn steel are hardly investigated up to now although the total elongation is higher than those of the TM and D-MMn steels [31].

  1. What was the step size for the EBSD measurement?

[Reply] “Step size” means “step distance” of the beam for the EBSD measurement. The smaller the step size, the higher the resolution. This parameter is a familiar one for the EBSD measurement. To deepen the understanding, we added the other parameters such as accelerating voltage, beam diameter, etc. to line 105.

  1. Some of the areas pointed out as the retained austenite seem to be unindexed points like Figure 3(C).

[Reply] YES. Some of the areas pointed out as the retained austenite are unindexed points because these maps are corresponding to α-Fe (BCC) phase. So, the authors added the following sentence to line 156.

In Figures 3a to 3c, the retained austenite looks like the black phase or points because of unindexed phases or points.

  1. This sentence ‘’ fMA and fθ are volume fractions of the MA phase and carbide ’’ should be removed from the Figure 3 caption as it is not related to this figure.

[Reply] The values of fMA and fθ are needed to deepen the understanding of the microstructure in Figure 3. So, we would like to keep the sentence in the Figure 3 caption.

  1. One of the critical steps to have high-performance Med-Mn steels is the intercritical annealing prior to the final treatment, why it has not been considered in this work?

[Reply] In this research, we deal with the third generation AHSS (Type B) with a tensile strength higher than 1.5MPa. As the D-MMn steel has a tensile strength of about 1 GPa (< 1.5GPa). The steel which is compared to the M-MMn steel is the TM steel (equivalent to the 1.5Mn steel) having tensile strength higher than 1.5 GPa. So, we excluded the D-MMn steel.  

  1. Considering the tensile properties, the mechanical properties are very similar to the maraging TRIP steels and far from the Med-Mn steels.

[Reply] The M-MMn steel is characterized by a matrix structure of fresh martensite, not annealed martensite. In addition, the M-MMn steel includes the metastable retained austenite, not reverted austenite. The microstructure easily brings on very high tensile strength. The microstructure of the D-MMn steel and the maraging TRIP steels are characterized by annealed martensite and a large amount of reverted austenite. So, the M-Mn steel resembles the TM steel and is not similar to the maraging TRIP steel.

  1. The effect of grain size (both bcc and fcc phase) on the achieved mechanical properties can add more clarification on the results obtained from the different processing routes and alloys which have been used in this work.

[Reply] In this research, the sizes of coarse martensite and retained austenite tend to decrease with increasing Mn content. As the effect of these sizes results from high Mn content, the (iii) was modified, as follows.

(iii)            Mn concentration or content which contributes to the solid solution hardening and influences the sizes of prior austenitic grain, coarse martensite and retained austenite, as well as the prior austenitic boundary strength.

As the coarse martensite and retained austenite size are not explained in Results, the following sentence was added to line 208.

The sizes of the coarse martensite and retained austenite phase tend to decrease with increasing Mn content.

  1. Others

Ref. 41 was added. Please check the sentence of line 637 and References.

That’s all

Round 2

Reviewer 1 Report

To ensure the reliability of the data, the author is requested to provide XRD pattern data. The author does not need to insert data into this manuscript, just provide the data in the reply.

Reviewer 3 Report

The current version of the paper is good enough to be accepted for publication. 

Author Response

The current version of the paper is good enough to be accepted for publication.  

[Reply] Thank you so much for your kind comments.

Round 3

Reviewer 1 Report

 Accept in present form.